# Relationship between Concrete Hole Shape and Meso-Crack Evolution Based on Stereology Theory and CT Scan under Compression

**DOI:** 10.3390/ma15165640

**Published:** 2022-08-16

**Authors:** Weihua Ding, Lin Zhu, Hu Li, Man Lei, Fan Yang, Junrong Qin, Aiguo Li

**Affiliations:** 1State Key Laboratory of Eco-Hydraulics in Northwest Arid Region, Xi’an University of Technology, Xi’an 710048, China; 2School of Civil Engineering and Architecture, Xi’an University of Technology, Xi’an 710048, China; 3Shaanxi Key Laboratory of Loess Mechanics and Engineering, Xi’an University of Technology, Xi’an 710048, China

**Keywords:** concrete, CT image, meso-damage, meso-crack, macro-crack, hole, stereology, roundness, partitioning

## Abstract

To achieve more accurate prediction of the potential failure location and to conduct a deeper analysis of the failure mechanism of concrete constructions, it is critical to probe the evolution process of internal meso-cracks that bear various intensities of load. While a computer Tomography (CT) test provides a non-destructive detection technique for obtaining the internal meso-damage state of concrete, traditional image processing and Digital Image Correlation (DIC) are ineffective in extracting meso-damage information from concrete CT images. On the other hand, by observing the shape change law of concrete’s internal holes under load, it is proposed to use the hole roundness and area fraction formula, developed based on the stereology principle and morphology, to characterize and predict the potential failure location. Four features particularly addressed include the CT image as a whole, image equal partitioning, crack and non-crack areas, and representative holes. The approach is to explore the variation law of critical hole shape parameters, especially the hole roundness under different loading stages, and analyze the relationship between the change in hole shapes and the final macro-crack positions. It is found that compared with the average area fraction, the average hole roundness value of cross section images is more sensitive to the change in stress. In both uniform partitioning and non-uniform partitioning, the average hole roundness value near the final macro-crack location exhibits an increase trend with the stress, while the smoothing effect caused by the hole roundness averaging always exists. Near the final macro-crack location, the roundness of each individual hole is positively associated with the stress, while away from the final macro-crack location such a relation may not be observed. This trend expounds the evolution process of meso-damage in concrete, and the finding can be used to predict the accurate locations of macro-cracks.

## 1. Introduction

Initial meso-damage, such as air voids and meso-cracks, are contained inside concrete structures at the beginning of the formation of concrete. Concrete structures are inevitably subjected to external loads from their own weight, earthquakes, and impact force in the service process, resulting in further accumulation of internal meso-damage, initiation of new meso-cracks, additional crack development, and even macro-damage [1]. Such damage puts the concrete structure safety at risk and reduces the service life.

From the macroscopic scale, concrete is a homogeneous material. From the mesoscopic scale, concrete is a heterogeneous material, composed of aggregate, mortar, and air voids. Meso-damage is the phenomenological expression of the deterioration of the material’s mechanical properties from the mesoscopic scale. Before the peak strength of concrete, internal cracks in concrete are referred to as meso-cracks and after the peak strength, internal cracks called macro-cracks. A meso-crack is used as the measure of meso-damage for concrete materials.

### 1.1. Existing Studies

Under the condition of load, the process of initiation, propagation, connection, and penetration of meso-cracks in concrete reproduces the entire realistic evolution process of meso-damage. Therefore, research on the meso-damage evolution of concrete is helpful for exploring the macroscopic mechanical characteristics and the meso-damage mechanism. The purpose of this research is to seek clues for the mix design of concrete materials in engineering, and to provide ideas and methods for studying the failure criteria of concrete under various loads.

In recent years, various non-destructive testing technologies (NDT), including Scanning Electron Microscopy (SEM) [2,3], acoustic emission [4,5], fluorescence microscopy [6,7], and CT [8,9] among others have been applied in the field of concrete damage detection.

To quantitatively describe the evolution of concrete internal meso-damage under load, it is necessary to ensure that the size of the specimen is large enough to reflect the macroscopic properties of concrete materials, and the internal meso-damage of concrete is observable. However, the methods mentioned above have some shortcomings in the study of meso-damage of concrete. For example, SEM and fluorescence microscopies have the characteristics of large magnification, small field of view and small sample, leading to the test results not to reflect the whole picture of the internal mesoscopic scale of concrete [10,11]. Acoustic emission method is used to observe the position and distribution of acoustic emission events in concrete, and to predict the initiation process of meso-cracks. However, the accuracy is relatively low [12]. Real-time scanning CT image including concrete mesoscopic structure can be obtained by coupling CT scan instrument and loading test system equipment, which is helpful to study the mesoscopic damage evolution process of concrete quantitatively [13,14].

To track the evolution process of a meso-crack, using a traditional image processing method has previously been tried, for example the gray mean method, difference CT image method, contour analysis method, isodensity segmentation analysis method, and gradient method [15]. However, the results of using these methods are not satisfactory in characterizing the meso-crack evolution process at the peak strength of concrete. These methods all calculate the CT value corresponding to the pixel point in the CT image. However, the CT value of the image at the same position in different loading stages does not noticeably change before reaching the peak strength. Hence, these methods are not to be suitable when the results of the meso-crack evolution before the peak strength of concrete are investigated [16,17].

### 1.2. Stereology Theory and Its Applications

In 1847, Delesse et al. first proposed the stereology in the study of ore deposit profiles [18]. Late in 1961, a complete system of stereology theory was developed by Elias et al. [19]. The original intention of the theory was to solve the problem of description and statistical measurement of two-dimensional structures. So far, stereology has been widely used in the fields of material science, image processing and biomedicine.

Stroeven et al. extracted parameters describing the geometric structure of cementitious materials based on stereological 2D and 3D particle shape evaluation methods and constructed a particle shape estimator based on these parameters [20]. Jutzeler et al. obtained the actual 3D volume fraction of clastic groups from 2D cross-sectional images of rock clastic sediments based on the principle of stereology [21]. Debnath et al. studied the structural characteristics of porous concrete based on stereology theory [22]. Peng applied the principle of stereology and CT to study the length-width ratio and roundness of irregular coarse-grained soil [23]. Ding et al. processed the electron microscopic images based on the quantitative stereology method, and studied the variation law of the internal pore structure of concrete containing different amounts of Super Absorbent Resin (SAP), and found the relationship between the average pore size of pores in concrete and the compressive strength of concrete with the SAP content [24]. Wu et al. applied the stereology method and X-ray CT to estimate the gradation of asphalt mixtures [25]. Analytical efficiency and accuracy of their models are superior to those of traditional pore structure analysis methods. The above studies demonstrated the effectiveness of the stereological method in quantitatively characterizing the morphological characteristics of materials.

### 1.3. Existing Problem

The meso-crack occurs usually near the peak strength of concrete when it is clearly active, but it is difficult to obtain CT images of concrete specimens because the peak strength of concrete cannot be accurately predicted. In addition, even if the CT image contains the newly generated meso-crack information, such a meso-crack is hardly going to appear intuitively due to its small size and imaging interference of aggregates and holes (holes have the same meaning with air voids in this paper, which is more intuitive and convenient).

This greatly increases the difficulty of extracting meso-damage information. To resolve this problem, DIC has been previously applied to the identification of meso-cracks in CT images. This method takes the sub-image composed of amount pixels as the basic unit, and it is used to identify the displacement according to the correlation coefficient between the sub-image and reference sub-image. Furthermore, the method is used to obtain the displacement field and strain field of the sub-image inside the loaded concrete, allowing meso-crack evolution process to be judged according to the strain gradient of the strain field.

However, this method has too many sub-volume pixels, and the smoothing effect seriously reduces the resolution of CT images. This leads to the unsatisfactory effect of DIC in identifying microscopic cracks in CT images.

### 1.4. Approach to Resolve the Problem

Aiming to solve the above problem, we begin with three facts about holes:Holes in concrete are easily to deform under load;Holes are clearly visible in the CT image;The deformation of holes is closely related to the evolution process of the meso-cracks, and relevant concepts can be developed based on stereology. The shape of the holes as well as the change in the hole shapes can be analyzed so that the active area of meso-cracks can be indirectly determined.

On the basis of the literature survey in Section 1.1, the existing studies focused on the static mesoscopic properties of unloaded concrete or rock materials, and to our best of knowledge, a lack of a sufficient research where the holes’ shape change process is investigated during the evolution process of a concrete meso-structure under load. Hence, a CT test of concrete under static pressure will be first carried out to obtain CT images that record the evolution process of concrete meso-damage. Then, image processing technology is used to identify, segment, and extract the meso-damage area inside the concrete material, and stereological parameters are determined to quantitatively characterize the concrete meso-crack. Furthermore, from four perspectives—overall image, equally partitioned images, crack and non-crack areas, and representative holes—the variation law of hole roundness under different loads will be observed, and the dependent relationship between concrete meso-damage and static pressure load is assessed in terms of the stereo parameters as bridges.

The rest of this paper is organized as follows. Section 2 presents the procedure for CT scanning test of concrete specimen. Section 3 introduces segmentation and quantitative shape characterization of holes and cracks in concrete CT images. In Section 4, the results will be carried out and analyzed. In Section 5 and Section 6, the results will be discussed, and conclusions will be drawn.

## 2. Research on Concrete Failure Process Based on CT Scanning Test

Meso-damage information can be obtained through concrete CT images, and this is the basis of our study.

### 2.1. Concrete Composition

Emeishan cement (moderate Heat Portland Cement, P•O 42.5), were obtained from Emeishan City, Sichuan Province, China. Guang’an I fly ash were obtained from Guang’an City, Sichuan Province, China. The chemical composition of cement was detected according to China National Standard GB175-2007, as shown in Table 1. The aggregate and sand are manually mined and screened granites. The size of coarse aggregates for the first-grade concrete is 10–20 mm, and the compressive strength of standard cube is 34.83 MPa. Table 2 gives the mix proportion of concrete.

### 2.2. Concrete CT Test System and Test Parameters

The concrete CT test system is composed of scanning equipment and a loading system, as shown in Figure 1. The CT scanning equipment used in this study is PHILIPS 16-slice spiral CT (manufactured by ROYAL PHILIPS in Amsterdam, Netherlands) at Renhe Hospital, Three Gorges University in China. CT image size is 512 × 512. The scanning voltage is 140 kV and the electric current is 200 mA. The spatial resolution is 24 lp/cm and the scanning thickness is 2.5 mm. The loading device is a portable loading device independently developed by Xi’an University of Technology, China. The test adopted a first-grade concrete cylinder specimen with a diameter of 60 mm and a height of 120 mm.

### 2.3. Test Procedure

The specific test procedure of concrete CT test follows the following steps:Install the specimen, and scan the specimen for the first time before applying load, and obtain the CT image of the concrete specimen in the initial state.Set 0.5 kN/s as the load application rate and stop loading when the load reaches 30 kN. Perform the second scan.To prevent any large displacement because of a sudden fracture of the specimen caused by the load control, the load control was changed to displacement control, with loading rate 0.005 mm/s and a target displacement of 0.70 mm. Perform the third scan while keeping displacement unchanged.Set target displacement as 0.85 mm and perform the fourth scan at the target displacement.Set the target displacement as 1.00 mm. However, when the displacement reaches 0.92 mm, the specimen breaks, and the static pressure load rapidly drops to 26.43 kN, and the fifth scan is performed at this load.

Loading rate in this study is in line with the China standard (Test code for hydraulic concrete, SL/T 352-2020), and this is consistent with the ISO standard (Testing of concrete-Part 4: Strength of hardened concrete, 1920-4-2020).

According to the displacement and load values of each scanning stage recorded during the test, a stress-strain curve is drawn, as shown in Figure 2a and explained below.

### 2.4. Analysis of CT Test Results

Under the uniaxial static pressure, the cylinder specimen generally shows an upright “v” and inverted “v” combined shape failure, which is mainly caused by the effect of the end friction effect, i.e., the structure effect of the specimen. The specific shape and formation process of the combined shape failure, such as the location of the crack point, the propagation direction of the meso-crack, and the connection mode, reflect the material properties. The combined shape failure mode is a comprehensive effect result from the stress distribution inside the specimen, the strength and deformation of aggregate, mortar, and the characteristics of the joint surface between aggregate and mortar. The meso-crack evolution process of concrete can reflect the meso-mechanical properties of concrete materials and determine the macroscopic strength and deformation characteristics.

Figure 2a shows the Stress-Strain curve of the specimen, and Figure 2b present cross-sectional CT images corresponding to the five scanning stages at different heights of the concrete specimen81 cross-sectional CT images can be obtained in one scan, and Figure 2b only displays representative cross-sectional CT images at scan horizons 70, 150, and 230.

In Figure 2 one can see that the process of crack initiation, propagation, and connection in concrete cannot be directly observed in the CT images before the peak strength, i.e., the evolution process of the meso-damage inside the concrete specimen cannot be judged by the naked eyes before the failure.

An obvious macro-crack appeared on the CT image obtained from the fifth scan, at which point the concrete specimen has damaged. This means that from the material viewpoint, the aggregates, mortar, and holes are clearly visible. CT images provide an effective method to observe the internal structure of concrete, and the resolution can meet the observation needs. However, from the perspective of meso-damage evolution of concrete materials, the resolution of the CT images cannot meet the observation needs.

Therefore, it is indispensable to seek appropriate image processing methods to mine the information of meso-crack initiation, propagation, connection, and penetration contained in CT images. In the next section, the methods for extracting the meso-crack information from the CT images of the concrete specimen will be focused on at each scanning stage before concrete failure so that the evolution law of the meso-damage before the peak strength of load will be revealed.

## 3. Segmentation and Quantitative Shape Characterization of Holes and Cracks in Concrete CT Images

To actualize the study of the hole shape, two aspects of the work must be carried out. One is to separate the holes and cracks in CT images, and the other is to find appropriate parameters to describe the hole and crack shapes for obtaining their shape change information of meso-damage at different stress stages.

### 3.1. Calculation Principle of Quantitative Parameters of Hole Shape Based on Stereology Theory and Image Segmentation Method

In stereology, the quantitative description of 3D structural features of materials is achieved with a 2D projection. Figure 3 shows the 2D projection of the 3D crack. Based on stereology theory, the basic quantitative parameters are length, width, area, and perimeter of the 2D crack. The length and width were defined by the bounding-box of cracks, as shown in Figure 3. Then, the area fraction ($A_{AC}$), roundness (R) and roundness mean ($\bar{R}$) are defined, which can be calculated according to Formulas (1)–(3). ${\bar{A}}_{AC}$ refers to the ratio of the damaged area and the entire cross-sectional area of specimen, which is one of the important indicators for evaluating the degree of concrete damage. *R* indicates the degree of similarity between a particle and a circle. This indicator can be used to assess whether or not a hole or crack is close to a circular shape. A larger value of *R* means more branches of the hole or crack, thinner and longer contours of the crack, and more irregularly. The roundness of any circle is 1, which defines the critical value. The roundness of all other enclosed shapes is greater than 1.
(1)$${\bar{A}}_{AC}=\frac{A_{AC}}{S}$$
(2)$$R=\frac{P_{AC}^{2}}{4\pi A_{AC}}$$
(3)$$\bar{R}=\frac{\sum_{i=1}^{N}R_{i}}{n}$$

In Formulas (1)–(3), *A_AC_* is the total area of all holes and cracks on the scanning section, m^2^; *S* is the area of the entire scanning section, m^2^; *P_AC_* is the perimeter of each hole or crack in the scanning section, m; *n* is the total number of the holes and cracks in any binary image.

### 3.2. Determination of the Segmentation Method of Holes and Cracks in Concrete CT Images

To obtain the appropriate segmentation method of holes and cracks in CT images, three threshold segmentation methods including the minimum cross-entropy thresholding method [26], fuzzy threshold [27] and minimum error threshold [28] are studied in this research. Figure 4 displays the original CT image and segmentation results of holes and cracks.

A comparative study was preformed to choose the appropriate image segmentation method, as shown in Figure 4a. The red ellipse highlights the segmentation results of three image segmentation methods on the same area in original CT image. Compared to the Figure 4a, it can be seen that the cracks and holes in the original CT image are well segmented using the minimum cross-entropy thresholding segmentation method, and the boundaries of holes and cracks are clear. The minimum cross-entropy thresholding segmentation method is adopted to segment holes and meso-cracks in concrete CT images by comparing the results of the three segmentation methods.

### 3.3. Extraction and Marking of Holes and Cracks

The original cross-sectional CT image (Figure 5a) was segmented based on the minimum cross-entropy thresholding method, and a binary image of holes and cracks was obtained (Figure 5b). During CT imaging, since the sampling interval cannot be too close and the densities of air, aggregate and mortar vary largely, the edge effect will occur, causing the CT numbers of the edge parts of the specimen to be inaccurate. To avoid the influence of such edge effects on the calculation results, the marked red area is cropped in the binary image of holes and cracks, and the image outside the red area is removed. The result is shown in Figure 5c. The holes and cracks are colored to highlight the shape, size, and concavo-convexity of individual holes and cracks, and the results are shown in Figure 5d. In order to be identified easily when individual roundness of holes and cracks is targeted, they are highlighted with different colors (Figure 5d). The explanation for colors is suitable to Figure 9 and Figure 11.

To obtain the essential quantitative information, such as number, length, width, and perimeter of holes and cracks, a program first is compiled to process each hole and each crack in a loop to obtain the position and number of closed holes and cracks, and the holes and cracks are numbered. Finally, the parameters including the length, width, perimeter, and area of each separated connected area in the image are calculated, and the roundness and area fraction of holes and cracks are computed using these parameters.

## 4. Results and Analysis

### 4.1. General Variation Law of Shape Parameters of Holes and Cracks in the Full Cross-Sectional CT Images with the Stress

According to Formulas (2) and (3), using MATLAB, the average roundness and area fraction of all holes and cracks are calculated from the binary cross-sectional images taken at different heights of the specimen and under different loading. Figure 6a,b give the distributions of the average roundness and the area fraction along the height of the specimen, respectively.

From Figure 6a, it can be seen that before the load is applied (scan stage 1), the average roundness changed randomly along the height direction of the specimen, and the variation range is large because of the initial damage discreteness in concrete material. Before the specimen reaches the peak load (scan stages 2, 3, 4), the average roundness at different heights of the specimen at the three loading stages changed to different degrees with positive and negative increments. Hence, before the peak strength, the change in the average roundness does not show an obvious connection to the meso-cracks. In the fifth stage, the specimen was completely destroyed at the post-peak strength stage, and the average roundness changes greatly in each scanning section. The reason for this is that the newly appeared larger cracks are involved in the calculation, and the maximum roundness position corresponds to where the maximum damage occurs in the specimen.

Figure 6b shows that before no load is applied (scan stage 1), the area fraction reaches the maximum value at the layer numbered 143. The reason for this is that the area occupied by the holes in this layer is larger. The changes in area fractions at other scan layers are smaller, reflecting that the inhomogeneity of the damage in the initial stage of concrete.

The area fractions of the second, third and fourth scan stages are consistent with the area fraction of the first scan stage, with some little change only, indicating that before peak strength all internal holes’ areas do not change significantly. In the fifth scanning stage, the specimen was completely damaged, and this belongs to the post-peak strength stage. The change in the average area fraction at each scanning layer showed obvious differentiation and reached the maximum value at the layer 143. This is because the area occupied by the newly grown cracks in this layer is larger.

Comparing and analyzing Figure 6a,b, it is noticed that before the peak load, the average roundness changes irregularly and slightly with the increase of the load, but the area fraction basically does not change while the load increases. This phenomenon reveals two findings: First, the change in load causes the change in hole shape, but it has little effect on the change in the proportion of hole area. In concrete under compression, the internal holes of concrete are prone to deformation, and the study of roundness parameter is more meaningful. The second finding is that the average roundness and area fraction have no obvious effect on predicting the meso-damage evolution process. This is because of the smoothing effect, resulted in when the average roundness and area fraction are obtained by statistical calculation of all holes and cracks in each CT image. In the next, the variation law on roundness average value of holes during meso-damage evolution process in concrete by image partitioning will be studied.

### 4.2. Study on the Variation Law Regarding the Average Roundness of Concrete Holes and Cracks with Stress under Different Area Division Conditions

#### 4.2.1. Variation Law on the Average Roundness of Holes and Cracks with Stress in Equal Divisions

In this section, the CT images of concrete are divided into equal sections, and the average roundness of holes and cracks at all scanning layers is calculated to study the variation law of meso-cracks in concrete before macro-cracks occur. Figure 7 shows the specific partitioning principle. The circular image is divided into three equal parts along the diameter AB, and they are marked as the left area (L), middle area (M), and right area (R).

Six cross-sectional CT images (S120, S130, S140, S150, S160, and S170) were uniformly selected along the height direction of the concrete specimen (Figure 2), and for the holes and cracks in the L, M and R areas of the six cross-sectional CT images, their respective average values of roundness were calculated at the five scanning stages. The average value of roundness is shown in Figure 8a–f.

By analyzing Figure 8a–f, we conclude that before the peak strength, at the selected six sections, a change pattern for the average roundness of the holes and cracks in the L and M zones is not observed: here are increases and also decreases. The average value of hole and crack roundness in the R zone at the selected six sections shows an overall trend of increase as the load increases, and in the R zone, the law is more pronounced at the lower sections of the specimen. Clearly, this means that the meso-crack propagates upwards from the lower part of the specimen. Combining with Figure 2b, the main crack appeared on the right side of the specimen after the failure of the specimen, namely the area where the R zone is located. Therefore, the meso-crack activity in R zone is stronger, and hence the average value of hole and crack roundness in R zone is calculated separately to avoid the interference of the other two zones. Therefore, the equal division study based on the average roundness can help predict the location of macro-crack better than the average roundness of the whole section.

However, the average roundness of holes and cracks in equal divisions of the image is still difficult to reflect the local law of shape change in each individual hole or crack, which is not conducive to predicting the specific and accurate location of meso-cracks.

#### 4.2.2. Variation Law of the Average Roundness of Holes and Cracks with Stress in the Cracked and Non-Cracked Regions

To predict the location of macro-cracks more accurately, the concrete image is divided into either a crack zone or non-cracked zone in this section. The specific partition principle is: (1) Crack zone: this zone is near the macroscopic cracks at the 5th scanning stage (failure stage). (2) Non-cracked zone: this zone is far from the macro-cracks in all five scanning stages.

Based on the above zoning principle and former same positions, 30 CT images of the five scanning stages at these six positions were analyzed. The cracked area and the non-cracked zone of the image are divided, and the division result is shown in Figure 9. The average roundness of holes and cracks in the cracked area and the non-cracked zone in 30 images, respectively, was calculated, and the variation law of the average roundness of the cracked area and the non-cracked zone in the CT images at six positions with respect to different stress was obtained, as shown in Figure 10.

From the analysis of Figure 10, it can be seen that before the peak strength, the average roundness of the holes and cracks in the non-cracked zone, except for the S170 horizon, exhibits basically a downward trend, indicating that there are no obvious signs of meso-crack activity around the holes.

In the cracked zone, before the peak strength, the average roundness of the holes at the S120 layer decreased slightly. The average increase in the roundness of holes and cracks at the S130 and S140 layers is small. The average roundness of the holes and cracks at the S150 layer has begun to increase, and the downward trend has been suppressed. The average increase in the roundness of holes and cracks at the S160 and S170 layers is significant, and this indicates that the meso-cracks around the holes develop early, eventually leading to the generation of macro-cracks. It shows that the propagation direction of the meso-crack is from the lower part of the specimen upward, which is consistent with Figure 8. Because the study area of the holes and cracks has changed, the average change trend of the roundness in Figure 10 is more obvious than that in Figure 8.

In summary, before the failure of the concrete specimen, the average roundness of holes and cracks in the future crack area is monotonically increasing compared with respect to the area without cracks in the future. This means that the average roundness is helpful in predicting future crack areas, in contrast to the efforts using various image processing methods that still cannot help locate meso-cracks before peak strength.

Since the initiation of meso-cracks is a process in which cracks occur at one or several points first and then gradually expand, the average roundness of holes and cracks divided into crack and non-cracked zone in the image still has a smoothing effect. The law is not conducive to predicting the specific and precise location of meso-cracks.

#### 4.2.3. Variation Law of Representative Holes’ Roundness with Stress

Concrete CT images contain clear information of the shape and distribution of aggregates, mortar, and holes in concrete. Because the size of initial crack in concrete is small, initial crack cannot be seen in CT image usually. In this section, holes in concrete are considered only. When extracting the roundness parameters of holes, the shape parameter information is gained by taking the average of roundness parameter values of individual holes. Consequently, the effective information about the shape change in each individual hole is suppressed. In this section, the change pattern of the roundness for individual hole under each loading stage will be addressed. For such a purpose, two sections were taken in each of the upper (S70~S120), middle (S130~S170), and lower (S180~S220) layers, and the representative hole numbers were labeled as A, B, C, D, E, and F, respectively (Figure 11).

Figure 12 shows the variation law of the roundness of these six holes A to F with the scanning stage.

It can be seen from Figure 12 that the roundness of the six representative holes in different layers varies with load. The roundness of holes A, B, and E increases continuously with the rise of the load. In particular, the roundness increases significantly in the two loading stages before approaching the peak strength, indicating that the meso-cracks around the holes are active, and finally macro-cracks near the holes A, B, and E appear. Holes C, D, and F are far away from the final macro-crack, and their roundness increases and decreases with the rise of the load. Hence, the load causes the hole roundness in the concrete to change correspondingly, but the change rules are different.

For an individual hole, monotonically increasing the trend of the roundness parameter is an indication of meso-crack activity and thus such information can be used to predict the location of macro-crack. This is more targeted than that predicted by the average roundness of holes in a specific area because it is no longer affected by the averaging effect.

## 5. Discussion

In Section 4, one can see that for the average roundness of holes and cracks, as the targeting area becomes smaller and smaller, and the stress increases before the peak strength of concrete, especially near the peak strength, the hole roundness shows a clear monotonous increasing trend. After the macroscopic rupture of the concrete sample has occurred, the concrete material enters the stage of strong softening, and a large number of meso-cracks are generated around the hole, which causes the shape and area of the hole to change significantly, and the roundness of the hole suddenly increases.

As the stress increases to cause the deformation of the hole, brittle fracture at the mesoscopic level occurs at the periphery of the hole near the final macroscopic crack, and the initiation of the meso-crack causes the hole shape to change and the roundness to increase. In the case that meso-cracks are difficult to observe in concrete CT images, the monotonous increase of the roundness of holes can indirectly reflect and predict the location of macro-crack. This, using roundness of individual holes to avoid the averaging effect, is more accurate in predicting locations of the final macroscopic cracks than using the average roundness for specific regions.

As far as the targeting area is concerned, since all the holes and cracks in the area are included, the average roundness is more reliable and stable, but at the same time the effective information of the shape change in a single hole is suppressed. Because the initiation of a meso-crack occurs at one or several points first and then gradually expands, only the shape of the holes near the initiation point of meso-crack changes, which is reflected in the roundness parameter. Because the area of a single hole is small and it is affected by the binarization segmentation method, the true area is not easy to measure, and the roundness value is not as reliable and stable as the average roundness of regional collective holes, which is the limitation of the proposed approach. In the applications, the individual roundness values and the average roundness complement each other.

It has much difficulty to obtain the series CT images including meso-crack initiation, propagation, connection, and penetration process of concrete with stress. This is because under most circumstances, meso-crack initiation occurs close to the peak strength of concrete, scan opportunity is difficult to grasp.

Finally, we put a note about further studies. To validate the proposed method in this research, more tests of the same type need to be carried out.

## 6. Conclusions

In this research, the CT test system is used to carry out the concrete static uniaxial compression test, and the CT images containing the meso-damage information of the concrete are obtained. Since the holes in concrete are prone to deformation after being stressed, the minimum cross entropy thresholding method is adopted to segment the holes and cracks in the cross-section CT image of the concrete. The idea is motivated from the stereology principle. The main conclusions of this paper are as follows:Before the peak strength of concrete, there is no obvious change rule for the average value of hole roundness and area fraction in the whole cross-section with the increase of stress, but the average value of hole roundness is more sensitive. The averaging of the hole and crack parameters seriously affects the characterization of the regularity in the shape change in individual holes with increasing stress.As the targeting area becomes smaller, it shows that the average roundness of the holes and cracks in the domain gradually change regularly with the increase of the load, and near the macro-cracks, the average roundness of holes has a monotonically increasing trend before the peak stress, suggesting that it is more valuable to analyze the variation law of roundness of individual hole.With the increase of stress, the roundness of individual hole near the location of the macro-crack has a monotonous increase trend, and the roundness of the hole far away from the macro crack has no such regularity, indicating that the roundness of individual holes before the peak strength increases monotonically. The meso-crack activity, which is difficult to be extracted from the image, has a definite correlation with the increase of the hole roundness, and the study of the roundness change in individual holes is helpful to predict the macroscopic crack positions.Since it is difficult for various image processing methods to display the new meso-cracks before the peak strength, the trend of monotonically increasing in holes’ roundness with the increase of stress is an important finding: such a trend prior to the meso-cracks can reflect the future positions of the macro-crack, and indirectly perceive the meso-crack before the peak strength. This is a breakthrough in the crack analysis in concrete CT images. In summary, it has important scientific research value and potential engineering application significance to carry out in-depth research on the change in hole shape in concrete.

## Figures and Tables

**Figure 1 materials-15-05640-f001:**
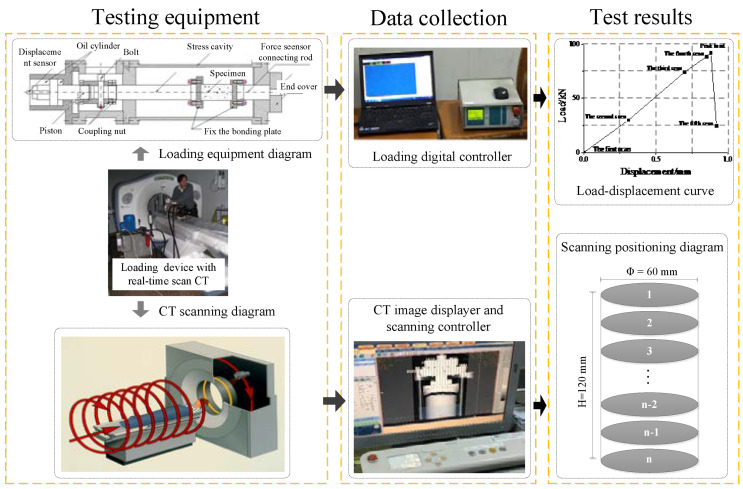
Portable loading device and scanner.

**Figure 2 materials-15-05640-f002:**
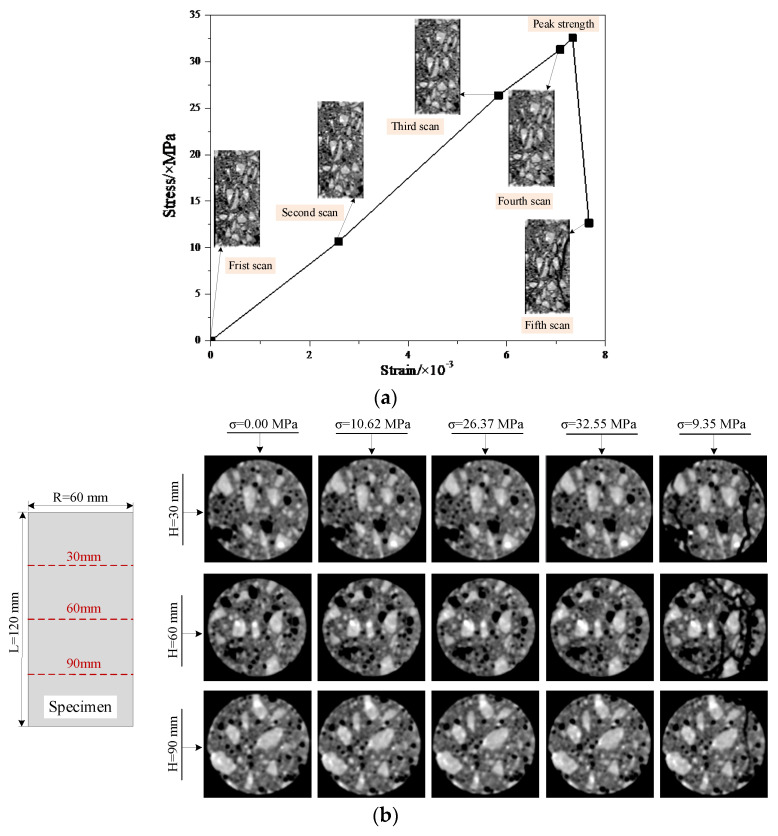
Experimental results of the concrete CT test, (**a**) Stress-strain curves of CONC-61 specimen, (**b**) Cross-sectional CT images of concrete specimens in different height at different stress stages.

**Figure 3 materials-15-05640-f003:**
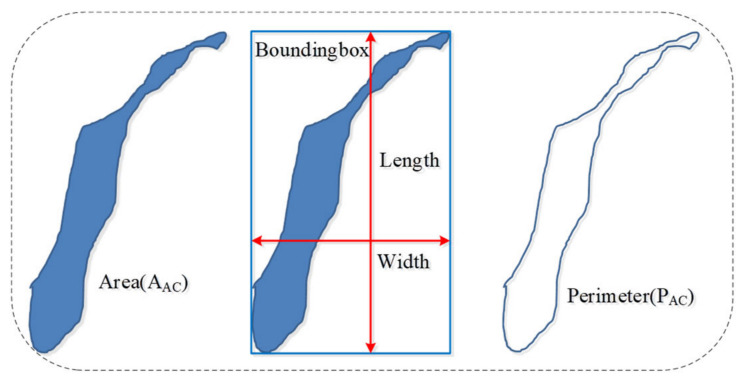
Hole or crack diagram.

**Figure 4 materials-15-05640-f004:**
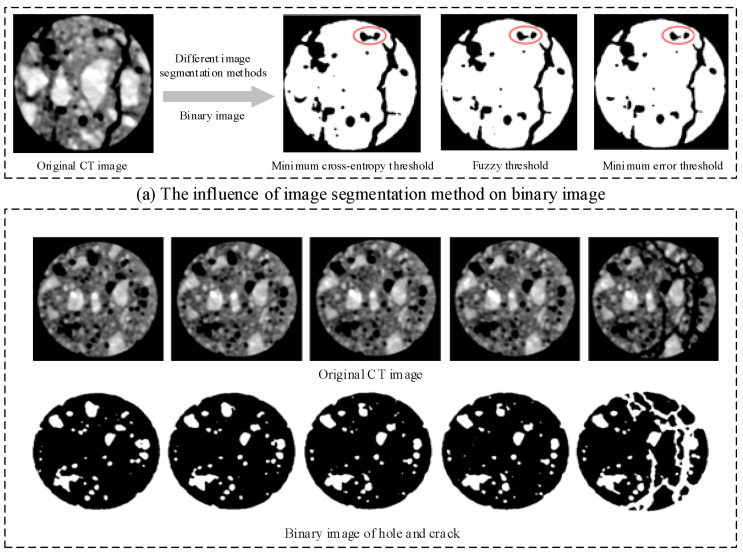
Determination of image segmentation method.

**Figure 5 materials-15-05640-f005:**
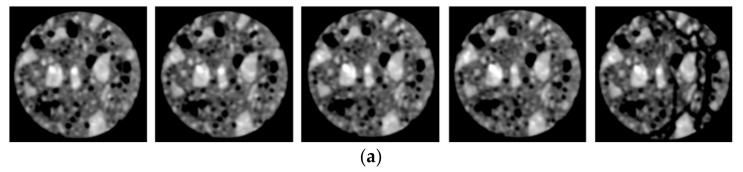
Crack and hole region extraction, cropping, colored for highlighingt, (**a**) Original CT images in the same slice at different scanning stages, (**b**) Binary images of holes and cracks, (**c**) Binary images of holes and cracks being cropped, (**d**) Holes’ and cracks’ calibration.

**Figure 6 materials-15-05640-f006:**
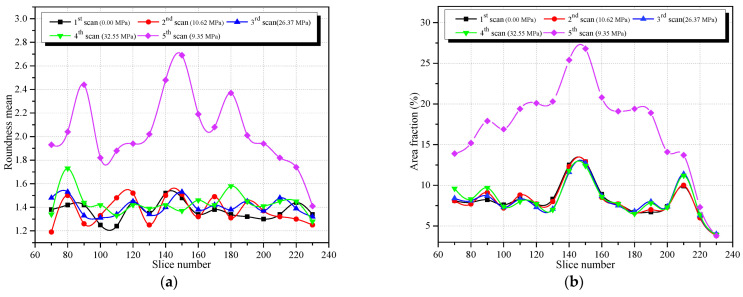
Changes in roundness mean and area fraction of holes and cracks along the height direction of the specimen at different scanning stages, (**a**) Roundness mean, (**b**) Area fraction.

**Figure 7 materials-15-05640-f007:**
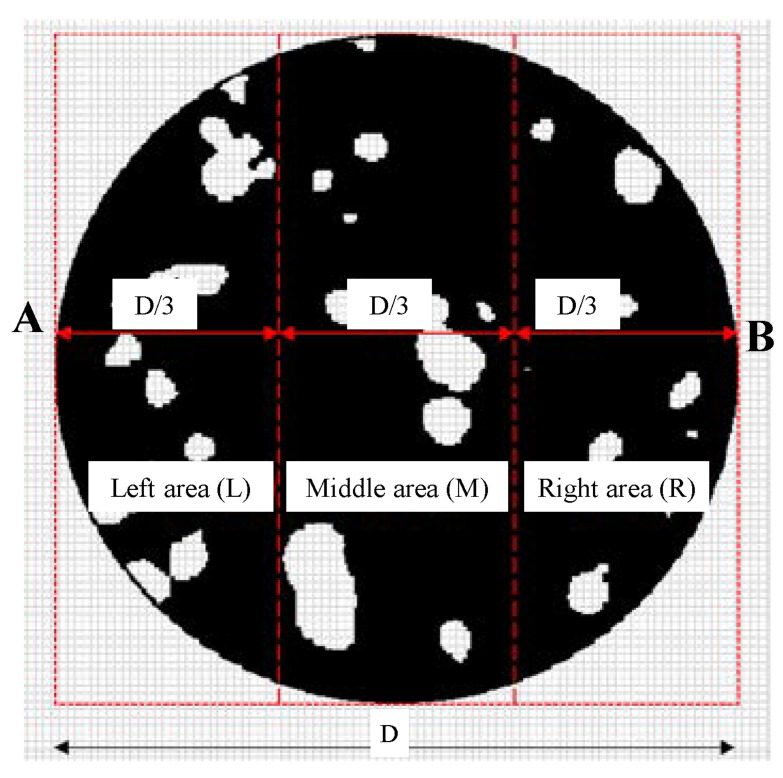
Cross-sectional zoning diagram.

**Figure 8 materials-15-05640-f008:**
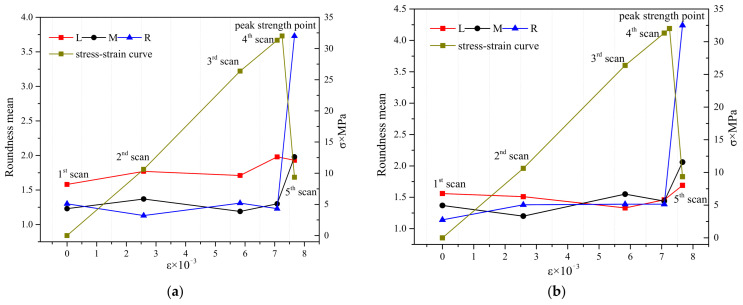
Roundness mean variation with loading in each area, (**a**) S120, (**b**) S130, (**c**) S140, (**d**) S150, (**e**) S160, (**f**) S170.

**Figure 9 materials-15-05640-f009:**
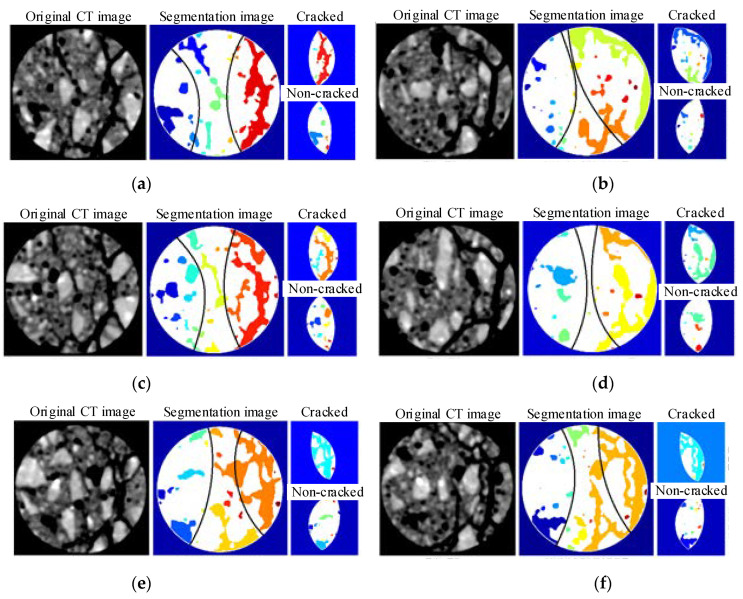
The partitioning results of cracked zone and no-cracked zone, (**a**) S120, (**b**) S130, (**c**) S140, (**d**) S150, (**e**) S160, (**f**) S170.

**Figure 10 materials-15-05640-f010:**
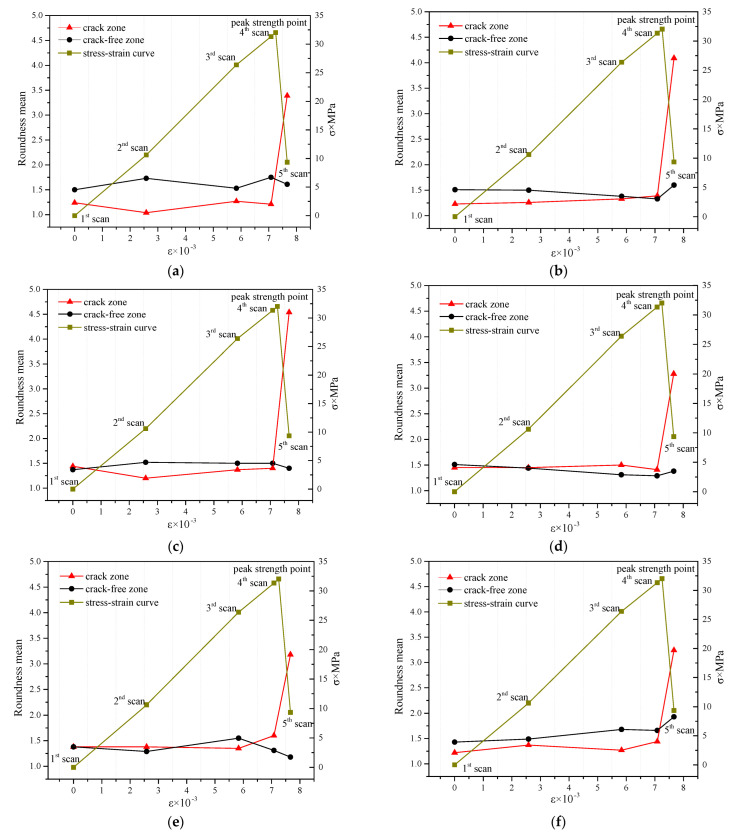
Variation of roundness mean at the two areas with loading, (**a**) S120, (**b**) S130, (**c**) S140, (**d**) S150, (**e**) S160, (**f**) S170.

**Figure 11 materials-15-05640-f011:**
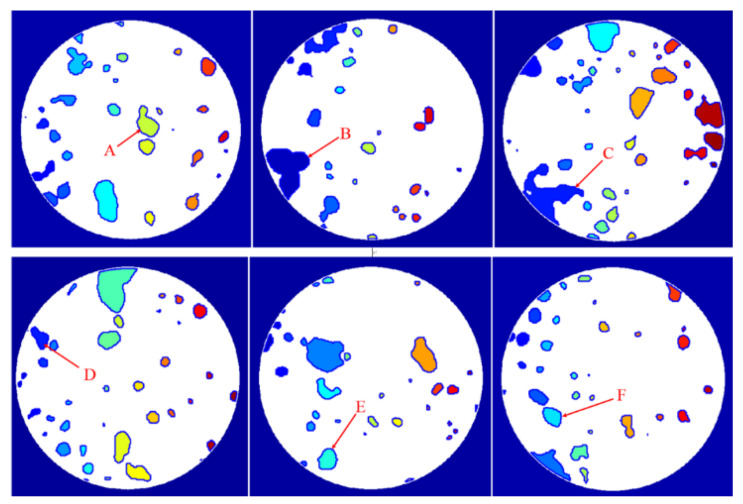
Diagram of hole location and number.

**Figure 12 materials-15-05640-f012:**
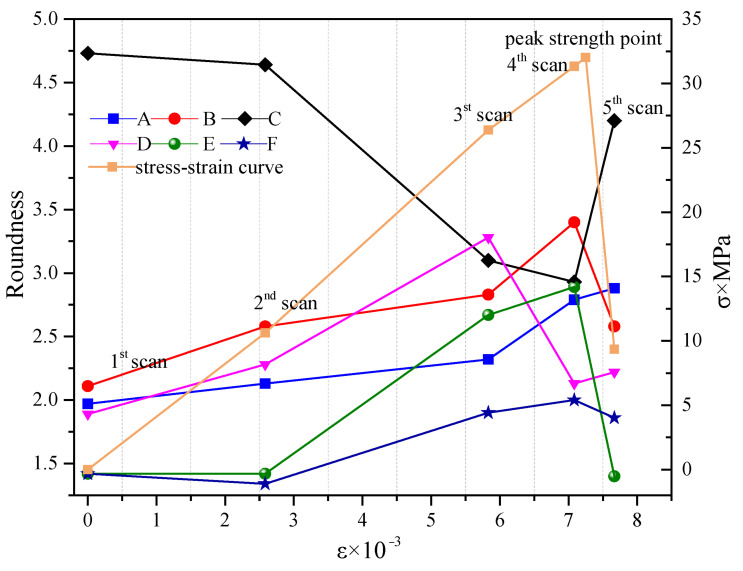
Variation on the roundness of each hole with the loading.

**Table 1 materials-15-05640-t001:** Chemical composition of cement (%).

SiO_2_	Al_2_O_3_	Fe_2_O_3_	CaO	MgO	SO_3_	f-CaO	R_2_O
21.47	3.55	4.88	60.84	4.44	2.72	0.30	0.33

**Table 2 materials-15-05640-t002:** Mix proportion of concrete (kg/m^3^).

Water	Cement	Fly Ash	Aggregate	Sand
260	459	196	803	1648

## Data Availability

Not applicable.

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
