# Peer review of "Relationship between Concrete Hole Shape and Meso-Crack Evolution Based on Stereology Theory and CT Scan under Compression"

_materials, 2022, doi:10.3390/ma15165640_

Round 1
Reviewer 1 Report
My comments in pdf file.

Author Response
The point-to-point reply for the comments:
Reviewers' comments:
Reviewer: 1
Comment 1. Introduction: “From the mesoscopic scale, concrete is a heterogeneous material composed of aggregate, mortar and holes."- change to "aggregate, mortar and air voids.” (line 45)
Response: Thank you for this correction. The word “holes” has been changed to “air voids” in the revised manuscript.
Comment 2. Explain more about the novelty of the present study.
Response: Thank you for this suggestion. According to the comments, the novelty of the present study was emphasized in the current manuscript. The trend of monotonically increasing in air voids’’ roundness with the increase of stress is an important finding since such a trend prior to the meso-cracks can reflect the future positions of the macro-crack, and indirectly perceive the meso-crack before the peak strength. This is a breakthrough in the crack analysis in concrete CT images. See line 507-511.
Comment 3. Section 2.1. Please add more information about concrete composition. What type of cement (Portland, slag cement etc.) did you use? Where the cement and ash were produced? The composition of concrete mixture and chemical properties of cement need to be presented in table. Please specify the standards of used materials.
Response: Thank you for this suggestion. According to the comments, the information about concrete composition has been added in the revised manuscript. The following has been added in the revised manuscript, See line 156-164.
The Emei Mountain cement (moderate Heat Portland Cement, P•O 42.5,) and Guang'an I fly ash were obtained from Sichuan Province, China. Chemical composition of cement was detected according to China National Standard GB175-2007, as shown in Table 1. Table 2 shows the mix proportion of concrete.
Table 1 Chemical composition of cement (%)
|
SiO2 |
Al2O3 |
Fe2O3 |
CaO |
MgO |
SO3 |
f-CaO |
R2O |
|
21.47 |
3.55 |
4.88 |
60.84 |
4.44 |
2.72 |
0.30 |
0.33 |
Table 2 Mix proportion of concrete (kg/m3)
|
Water |
Cement |
Fly ash |
Aggregate |
Sand |
|
260 |
459 |
196 |
803 |
1648 |
Comment 4. Section 2.3, line 84. Please give reference to ISO standard.
Response: Thank you for this suggestion. The ISO standard (ISO 1920-4-2020, Testing of concrete-Part 4: Strength of hardened concrete) has given in the current manuscript. See line 190-192.
Comment 5. Section 3.1: Add the unit for all components of Eq.1,2,3 and check carefully the symbols in lines 232-233.
Response: Thank you for this suggestion. According to the comments, the unit for all components of Eq.1,2,3 has been added in the revised manuscript. See line 255-257. The symbols have been checked carefully.
Comment 6. Section 3.3, Figure 5d. For a better representation of the results, a legend to the color scale should be available. For example, does the yellow color mean cracks?
Response: Thank you for this suggestion. According to the comments, the mean of colors in Figure 5d has been explained in the revised manuscript. In order to be identified easily when individual roundness of holes and cracks is targeted, they are highlighted with different colors. See line 282-284.
Comment 7. Make sure that all references and citations are exactly based on the style adopted by Journal.
Response: Thank you for this suggestion. According to the comments, all references and citations have been checked based on the style adopted by the Journal.
Reviewer 2 Report
The work is experimental and concerns the analysis of the structure of concrete under load. The aim of the study was to improve the prediction of the potential damage site and damage evolution. To analyze the damage, the computer tomography (CT) method was used to detect internal damage as an alternative to the DIC method allowing the observation of the outer surface.
The authors observed changes in the shape of the internal pores in the concrete after post-loading. The prediction of microcracks was based on the analysis of the variability of the pore shape parameters (roundness) at different stages of loading and the use of the relationship between the shape change and the location of microcracks. It has been shown that the mean value of the roundness of pores in the images of cross-sections is sensitive to changes in stresses.
Thus, the process of damage on a micro and macro scale has been explained, and the formulated dependencies will allow the damage prediction process.
The work is interesting from a cognitive point of view. The biggest drawback is the lack of clarification of some issues. However, my assessment is unequivocally positive. I present my detailed comments below:
1. Chapter 1: I suggest to emphasize the novelty of the proposed method.
2. Chapter 2.1: What aggregate fractions were used?
3. Chapter 2.2: The block diagram is difficult to read. Please increase the scale. I suggest marking macrocracks on the CT scans.
4. Chapter 3.1: Occurring physical quantities should be defined.
5. Chapter 3.3: The research description is very good. I have no comments.
6. Chapter 4.1: There is no reference to the order of the scans in relation to the stresses in the concrete in relation to the compressive strength.
7. Chapter 4.2: All results can be agreed with. I only have doubts about the omission of coarse aggregate in the scans.
8. Chapter 6: Please add directions for further work in this area.
Author Response
The point-to-point reply for the comments:
Reviewers' comments:
Reviewer:2
Comment 1. Chapter 1: I suggest to emphasize the novelty of the proposed method.
Response:
Response: Thank you for this suggestion. According to the comments, the novelty of the present study was emphasized in the current manuscript. The trend of monotonically increasing in air voids’’ roundness with the increase of stress is an important finding since such a trend prior to the meso-cracks can reflect the future positions of the macro-crack, and indirectly perceive the meso-crack before the peak strength. This is a breakthrough in the crack analysis in concrete CT images. See line 507-511.
Comment 2. Chapter 2.1: What aggregate fractions were used?
Response: Thank you for this suggestion. According to the comments, the mix proportion of concrete has been added in the current manuscript (See line 163), also shown below.
Table 2 Mix proportion of concrete (kg/m3)
|
Water |
Cement |
Fly ash |
Aggregate |
Sand |
|
260 |
459 |
196 |
803 |
1648 |
Comment 3. Chapter 2.2: The block diagram is difficult to read. Please increase the scale. I suggest marking macrocracks on the CT scans.
Response: Thank you for this suggestion. According to the comments, the block diagram has been redrawn. The new Figure 1 has been used to replace the old one in the revised manuscript (See line 173-174).
Comment 4. Chapter 3.1: Occurring physical quantities should be defined.
Response: Thank you for this suggestion. According to the comments, the physical quantities have been defined in the revised manuscript. The length and width were defined by the bounding-box of cracks, as shown in Figure 3.See line 243-249.
Comment 5. Chapter 3.3: The research description is very good. I have no comments.
Response: Thank you for your encouragement.
Comment 6. Chapter 4.1: There is no reference to the order of the scans in relation to the stresses in the concrete in relation to the compressive strength.
Response: Thank you for this suggestion. According to the comments, Chapter 4.1 has been revised. See Figure 6.
The main change is also provided below. Figure 2(a) shows the relationship between the order of the scans and stresses in the concrete. Combining Figures 2 and 6, we can know that the 1st scan corresponds to the stress of specimen subjected to 0 MPa, the 2st scan corresponds to the stress of specimen subjected to 10.62 MPa, and so on. The relationship between order of the scans and the stresses in the concrete was given in Figure 6.
Comment 7. All results can be agreed with. I only have doubts about the omission of coarse aggregate in the scans.
Response: Thank you for this suggestion. In Figure 5(a), the CT images clearly show the coarse aggregate, mortar and air voids in the concrete specimens at various stress stages. However, the coarse aggregate area in CT images is not segmented in this research, which is because the shape of coarse aggregate has little change with stress. See line 285.
Comment 8. Chapter 6: Please add directions for further work in this area.
Response: Thank you for this suggestion. According to the comments, the directions for further work in this area were added in the revised manuscript. In order to validate the proposed method in this research, more tests of the same type need to be carried out in the further studies. See line 479-480.
Reviewer 3 Report
The authors have used CT scanning image analysis technique to investigate the damage/cracking behavior in the concrete sample under static uniaxial compression test. The following comments should consider while revising the manuscript.
· The quality/visibility of Figure 1 must improve. Currently most of the text in the figure is not visible.
· In section 2.3 the test procedures, any reference are followed for the parameters such as applied load, loading rate, displacement, etc.?
· What are the limitations of the test method used here please highlight.
· Are the values shown in figures 8, 10, 12 the average values of all slices or just single value?
· Conclusion can be shortening by highlighting the major findings only. Currently, most of the sentences are too long.
Author Response
The point-to-point reply for the comments:
Reviewers' comments:
Reviewer:3
Comment 1. The quality/visibility of Figure 1 must improve. Currently most of the text in the figure is not visible.
Response: Thank you for this suggestion. According to the comments, the Figure 1 has been redrawn. The new Figure 1 has been used to replace the old one in the revised manuscript (See line 173-174).
Comment 2. In section 2.3 the test procedures, any reference are followed for the parameters such as applied load, loading rate, displacement, etc.?
Response: Thank you for this suggestion. According to the comments, the reference (Test code for hydraulic concrete (SL/T 352-2020, China)) and the ISO standard (ISO 1920-4-2020, Testing of concrete-Part 4: Strength of hardened concrete) have been added in the revised manuscript. See line 190-192.
Comment 3. What are the limitations of the test method used here please highlight.
Response: Thank you for this suggestion. According to the comments, the limitations of the test method has been added in the revised manuscript. It has much difficulty to obtain the series CT images including meso-crack initiation, propagation, connection and penetration process of concrete with stress. Because under most circumstances, meso-crack initiation occurs close to the peak strength of concrete, scan opportunity is difficult to grasp. See line 475-478.
Comment 4. Are the values shown in figures 8, 10, 12 the average values of all slices or just single value?
Response: Thank you for this suggestion. The average values are used in Figures 8 and 10 to represent the average roundness of all air voids in a selected slice (single slice). The roundness in Figure 12 is a single value describing the representative holes’ roundness (single hole).
Comment 5. Conclusion can be shortening by highlighting the major findings only. Currently, most of the sentences are too long.
Response: Thank you for this suggestion. According to the comments, the concluding section has been shortened to highlight the major findings, and longer sentences have been split into shorter ones.